# Application of Color Featuring and Deep Learning in Maize Plant Detection

**Haojie Liu [1], Hong Sun [1,*], Minzan Li [1] and Michihisa Iida [2]**

[1] Key Laboratory of Modern Precision Agriculture System Integration Research, Ministry of Education, China Agricultural University, Beijing 100083, China; hjliu267@163.com (H.L.); limz@cau.edu.cn (M.L.)

[2] Graduate School of Agriculture Kyoto University, Kyoto 606-8224, Japan; iida@elam.kais.kyoto-u.ac.jp

\* Correspondence: sunhong@cau.edu.cn; Tel.: +86-010-62737838

**Abstract:** Maize plant detection was conducted in this study with the goals of target fertilization and reduction of fertilization waste in weed spots and gaps between maize plants. The methods used included two types of color featuring and deep learning (DL). The four color indices used were excess green (ExG), excess red (ExR), ExG minus ExR, and the hue value from the HSV (hue, saturation, and value) color space, while the DL methods used were YOLOv3 and YOLOv3_tiny. For practical application, this study focused on performance comparison in detection accuracy, robustness to complex field conditions, and detection speed. Detection accuracy was evaluated by the resulting images, which were divided into three categories: true positive, false positive, and false negative. The robustness evaluation was performed by comparing the average intersection over union of each detection method across different sub–datasets—namely original subset, blur processing subset, increased brightness subset, and reduced brightness subset. The detection speed was evaluated by the indicator of frames per second. Results demonstrated that the DL methods outperformed the color index–based methods in detection accuracy and robustness to complex conditions, while they were inferior to color feature–based methods in detection speed. This research shows the application potential of deep learning technology in maize plant detection. Future efforts are needed to improve the detection speed for practical applications.

**Keywords:** precision agriculture; maize plant detection; target fertilization; color featuring; deep learning

## 1. Introduction

Site–specific fertilization (SSF), which is proposed within the framework of precision agriculture [1], aims to accurately apply fertilizer according to the spatial variability of crop and soil. This is of great significance in improving economic and environmental benefits by reducing inputs and environmental pollution [2–4]. Maize is a main crop with a wide row space. At the seeding stage, the canopy cover of maize is limited, there are gaps between the maize plants, and weeds can grow within the crop rows. When top–dressing is conducted, areas of soil and weeds, where fertilization input is not needed, will also be covered by blanket spraying systems, causing unnecessary inputs and waste. Target fertilization (TF) is proposed as an enhanced SSF strategy to meet the requirements during the seedling period. Specifically, the target spraying systems used in SSF can automatically apply fertilizer only in areas where maize plants (targets) are located, thus reducing waste in weed and soil background areas (non–targets) from blanket spraying. High target detection performance, which relies on the accurate and rapid identification and localization of maize plant targets against a background of weeds and soil, is the critical prerequisite

for TF application because it provides significant information for decision making in the subsequent TF operations [5,6]. However, the accurate detection of target plants is a challenge in practice, due to complex conditions such as changing light, vehicle vibration, and mixed weeds. Therefore, this study proposes a reliable method for maize plant detection, providing support for practical TF application.

Image processing technology is a promising approach for target detection tasks in agriculture [7–9] and can be summarized into two categories: threshold– and learning–based methods [10,11].

The principle of threshold–based methods is to group pixels into different categories by comparing the value of each pixel in grayscale images with one or multiple preset threshold values. During this process, grayscale images are generated by transformations from original images to accentuate regions of interest (ROIs) and attenuate undesired regions. The threshold values can be determined by different thresholding methods [12,13].

One widely used transformation is to calculate color indices from the original RGB (red, green, and blue) values. For example, the excess green (ExG) index provides a clear contrast between plant objects and soil background and has performed well in vegetation segmentation [14–16]. Based on the composition of the retina of the human eye—4% blue cones, 32% green cones, and 64% red cones—the excess red (ExR) index was introduced to separate leaf regions from the background [15]. Furthermore, the ExG minus ExR (ExGR) index was defined by subtracting ExR from ExG to combine their respective advantages [17]. Many other color indices, such as the normalized green–red difference index (NGRDI) [18] and modified ExG, have also been introduced [19]. Additional available color features can be obtained from the HSV (hue, saturation, and value) color spaces [20], which can be converted from the RGB color space. The determination of threshold values is the critical element of threshold–based methods. Although a fixed threshold can be proposed on the basis of empirical knowledge, this fixed value is only suitable for dedicated situations and cannot meet the requirements of lighting variations [21]. Dynamic adjustable threshold values are required to cope with complex field environments. Otsu's method is a leading automatic thresholding approach, whose criteria for determining a threshold is the minimization of the within–class variance and the maximization of between–class variance [22].

Grayscale images can be converted into binary images by a suitable threshold, with the result that vegetation pixels are distinguished from the soil background; thus, the vegetation segmentation process classifies the pixels into two classes: vegetation and non–vegetation. However, the vegetation class not only includes crops but also weeds. Therefore, a non–crop class removal process is necessary for TF application. Features such as shape [23], texture [24], and venation [25] are generally used to discriminate between different plants. However, the extraction of these features is a labor–intensive and time–consuming process, which depends on expert knowledge, and the selected features are not sufficiently robust to be used in all scenarios [13]. Therefore, the detection accuracy of threshold–based methods remains a challenge in practice due to complex conditions such as mixed weeds, changing light, and vehicle vibration [10,13]. As a result, research has begun to investigate learning–based methods to deal with complex field environments.

Learning–based methods classify pixels in an image into different categories according to the common properties of objects learnt by a machine learning algorithm. The learning–based algorithms for vegetation segmentation comprise the decision tree [26], random forest (RF) [27], Bayesian classifier [28], back propagation neural network [29], Fisher linear discriminant [30], and support vector machine [31]. Deep learning (DL), as a particular kind of machine learning, has gained momentum in various applications. DL is a powerful and flexible approach because it transforms data in a hierarchical way through several levels of abstraction [32]. Milioto et al. [33] proposed a convolutional neural network (CNN) to address the classification of sugar beet plants and weeds. Qiu et al. [34] used a mask region CNN (RCNN) to detect Fusarium head blight in wheat from color images. Tian et al. [35] used a YOLOv3–based model to realize apple detection and counting during different growth stages. Kamilaris et al. [36] conducted a survey of 40 DL–based studies relating to agricultural tasks such as leaf classification, plant detection,

and fruit counting. Moreover, a comparison of DL techniques with other existing popular techniques was also undertaken. Their findings indicated that DL performed better than other learning–based methods such as Support Vector Machines (SVM) and RF. The key advantage of DL is its capacity to create and extrapolate new features from raw input data, locating the important features itself through training, without the need for labor intensive feature extraction (FE) and expert knowledge [37]. Another advantage of DL is that it reduces the effort required for FE [38]; such effort occurs automatically in DL without the need for expert knowledge and professional skills [39]. Moreover, DL models have been proven to be more robust than other methods under challenging conditions such as changing illumination, occlusion, and variation [40,41].

Motivated by the advantages of DL mentioned in the literature, this study addressed the maize plant detection task for future TF application, using a DL–based method. Four color index–based methods were used as benchmarks to compare detection performance. According to the challenges described above, the performance evaluations involved the accuracy of identification and localization under different conditions. Furthermore, detection time was considered for practical TF application.

The DL method used in this study was YOLOv3, which is a state–of–the–art object detection DL algorithm designed for real–time processing. As a single–stage detector, YOLOv3 transforms the input image into a vector of scores and performs detection operations using a single CNN; thus, detection is generally faster than that of two–stage detectors, such as Faster–RCNN [42]. It has significant potential in agricultural detection tasks [43,44]. Furthermore, a faster version of YOLOv3—namely YOLOv3_tiny—has also been developed, using fewer convolution layers to promote real–time application with embedded computing devices, which have limited computing capability. It is reasonable that the faster speed of YOLOv3–tiny is achieved at the expense of reduced precision. Therefore, both YOLOv3 and YOLOv3_tiny were evaluated for maize plant detection in this study due to the pursuit of balance between speed and detection accuracy.

The color indices used in this study included ExG, ExR, and ExGR, which have been widely used by researchers as benchmarks for performance evaluation in their proposed methods [27,30,31]. The H component of the HSV color space was also included because it can provide a separate description of the object color in addition to the lighting conditions [20]. The subsequent thresholds were determined by Otsu's method, which is also a widely applied method [13].

## 2. Methodologies

### 2.1. Color Index–Based Methods

The scheme in Figure 1 illustrates the image processing procedure of color index–based methods. Firstly, the original RGB images taken in the field were transformed into grayscale images through color indices. Thereafter, a threshold was set to produce a binary image in which green pixels were highlighted against the soil background. Finally, the maize plant targets and non–targets were distinguished. The operational details of each procedure are demonstrated in the following subsections.

2.1.1. Color Index Computation

The original images were described in the RGB color space. For color index computation, the following normalization scheme was first applied to normalize the spectral components into the range of 0 to 1:

$$R^* = \frac{R}{R_{max}}, \ G^* = \frac{G}{G_{max}}, \ B^* = \frac{B}{B_{max}}, \tag{1}$$

$$r = \frac{R^*}{(R^* + G^* + B^*)}, \quad g = \frac{G^*}{(R^* + G^* + B^*)}, \quad b = \frac{B^*}{(R^* + G^* + B^*)}, \tag{2}$$

where R, G, and B are the actual pixel values from the original images based on each channel and $R_{max} = G_{max} = B_{max} = 255$; r, g, and b are the normalized pixel values. Then, the color indices employed in this study, namely the ExG, ExR, ExGR, and H values, can be computed as follows:

$$ExG = 2g - r - b, \tag{3}$$

$$ExR = 1.4r - g, \tag{4}$$

$$ExGR = ExG - ExR, \tag{5}$$

$$M = \max(R^*, G^*, B^*), \quad m = \min(R^*, G^*, B^*), \quad C = M - m, \tag{6}$$

$$H = \begin{cases} 0^0 & \text{if } C = 0 \\ 60^\circ \times \left(\frac{G^* - B^*}{C}\right) & \text{if } M = R' \text{ and } G^* \geq B^* \\ 60^\circ \times \left(\frac{G^* - B^*}{C} + 6\right) & \text{if } M = R' \text{ and } G^* < B^* \\ 60^\circ \times \left(\frac{B^* - R^*}{C} + 2\right) & \text{if } M = G' \\ 60^\circ \times \left(\frac{R^* - G^*}{C} + 4\right) & \text{if } M = B' \end{cases}, \tag{7}$$

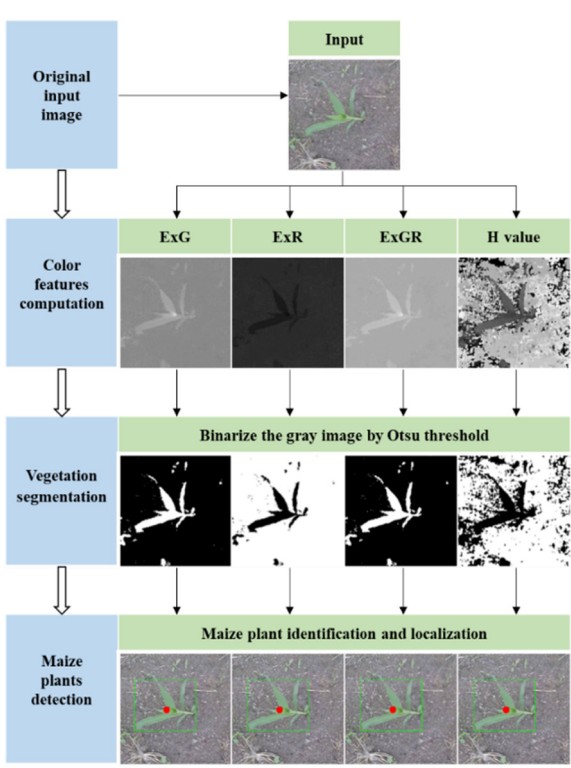

**Figure 1.** General workflow of color feature–based methods.

### 2.1.2. Otsu Thresholding

The thresholding task was formulated as a problem of dividing image pixels into two classes. Thresholding to obtain binary images was performed using Otsu's method [22]. This method involves iterating through all the possible threshold values to find the value that maximizes the between–class

variance while also minimizing the within–class variance of the image. Beginning with a random value, t, all pixels below the value of t belong to class $C_1$, while all pixels equal to or exceeding t belong to the class $C_2$. The probability of occurrence for each class is defined as

$$\omega_1 = \sum_{i=1}^{t} p_i, \tag{8}$$

$$\omega_2 = \sum_{i=t+1}^{max} p_i, \tag{9}$$

where $\omega_1$ and $\omega_2$ are the overall probabilities of classes $C_1$ and $C_2$, respectively; max is the maximum value in grayscale images; $p_i$ is the probability of intensity $i$, which is defined as

$$p_i = \frac{n_i}{N}, \tag{10}$$

where $n_i$ is the histogram count for pixel value $i$, and $N$ is the number of image pixels. The class means, $\mu_1$ and $\mu_2$, are the mean values of the pixels in $C_1$ and $C_2$, respectively, and are given by

$$\mu_1 = \sum_{i=1}^{t} ip_i/\omega_1 \tag{11}$$

$$\mu_2 = \sum_{i=t+1}^{max} ip_i/\omega_2 \tag{12}$$

The class variances, $\sigma_1^2$ and $\sigma_2^2$, are:

$$\sigma_1^2 = \sum_{i=1}^{t} (i-\mu_1)^2 p_i/\omega_1, \tag{13}$$

$$\sigma_2^2 = \sum_{i=t+1}^{max} (i-\mu_2)^2 p_i/\omega_2, \tag{14}$$

The between–class variance is given by

$$\sigma_B^2 = \omega_1\omega_2(\mu_1-\mu_2)^2, \tag{15}$$

The within–class variance is given by

$$\sigma_w^2 = \omega_1\sigma_1^2 + \omega_2\sigma_2^2, \tag{16}$$

### 2.1.3. Maize Plant Discrimination

The vegetation segmentation process classifies pixels into two classes: vegetation and non–vegetation. Both maize plant targets and weed objects are included in the vegetation class. It is thus necessary to further distinguish maize plants from weeds for the TF application. Though features such as shape, texture, and venation are generally employed to distinguish different plants, the extraction of these features is usually laborious and depends on expert knowledge. These features are also limited to dedicated scenarios. Compared with these features, size is an easily extracted feature. Furthermore, maize plants are the main objects in gathered images, and the size of maize plants is often larger than that of weeds in fields. Thus, size–based filters were applied to distinguish maize and weed plants in the study. All the connected

domains in a binary image were compared with a size threshold. Domains with a size greater than the size threshold were identified as maize plant domains, while others were identified as weed domains. The threshold was determined as 40% of the size of the input image to ensure that it could dynamically adapt to input images of different sizes. After size filtering, the minimum circumscribed rectangle of the remaining domain was computed as the bounding box, and the centroid of the rectangle was computed to represent the position of the target maize plant in an image.

## 2.2. DL Methods

### 2.2.1. Network Architectures

As shown in Figure 2, the architecture of YOLOv3 has 75 convolutional layers. It consists of residual blocks and concatenate operations, which add the supplemental "residual" to the previous features in the model and simplifies learning complexity. With this design, the network is more capable of capturing the low– and high–level features. As a fully convolutional network, YOLOv3 is able to handle images of any size because of the absence of a fully connected layer in the network, and it can precisely generate feature maps at three different scales by down–sampling the dimensions of the input image by 32, 16, and 8. Thus, for a $416 \times 416$ image, the resultant feature maps would have sizes of $13 \times 13$, $26 \times 26$ and $52 \times 52$. YOLOv3–tiny is a simplified version of YOLOv3, with 13 convolutional layers. The architecture of YOLOv3–tiny is shown in Figure 3. The network generates feature maps at two different scales, which are acquired by down–sampling the dimensions of the input image by 32 and 16. Both YOLOv3 and YOLOv3–tiny realize final maize plant detection using a similar means of regression, as detailed in the following subsection.

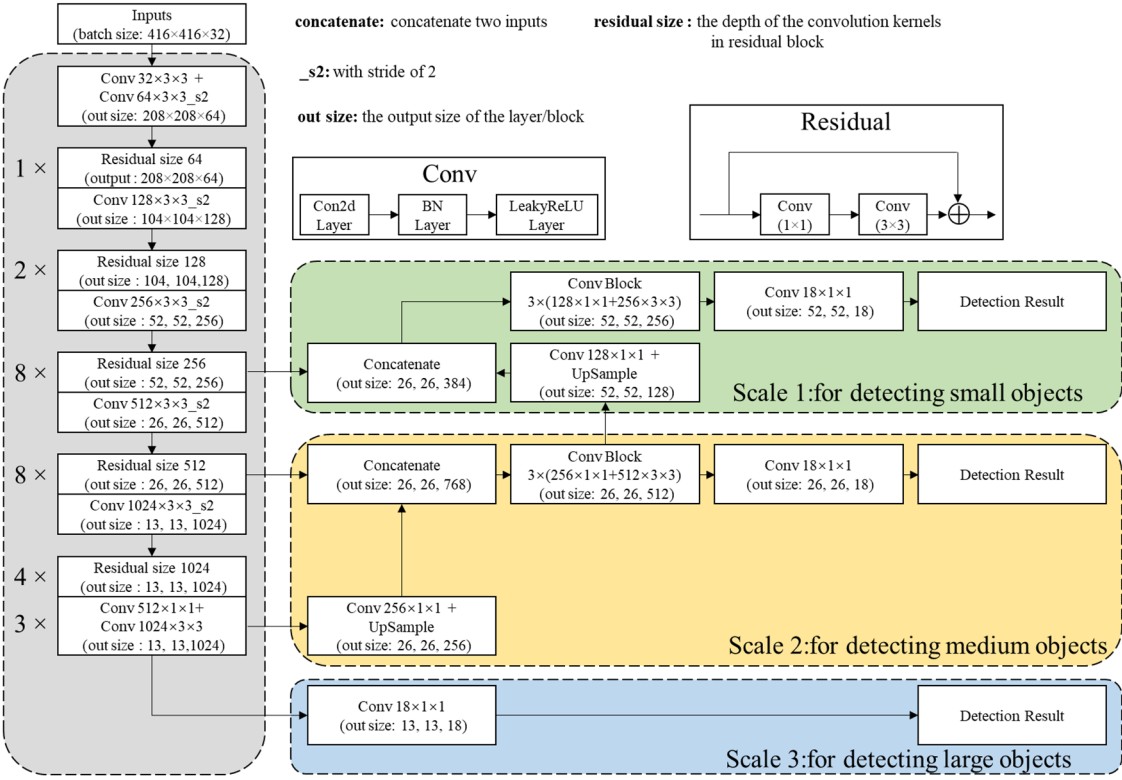

**Figure 2.** YOLOv3 detection architecture.

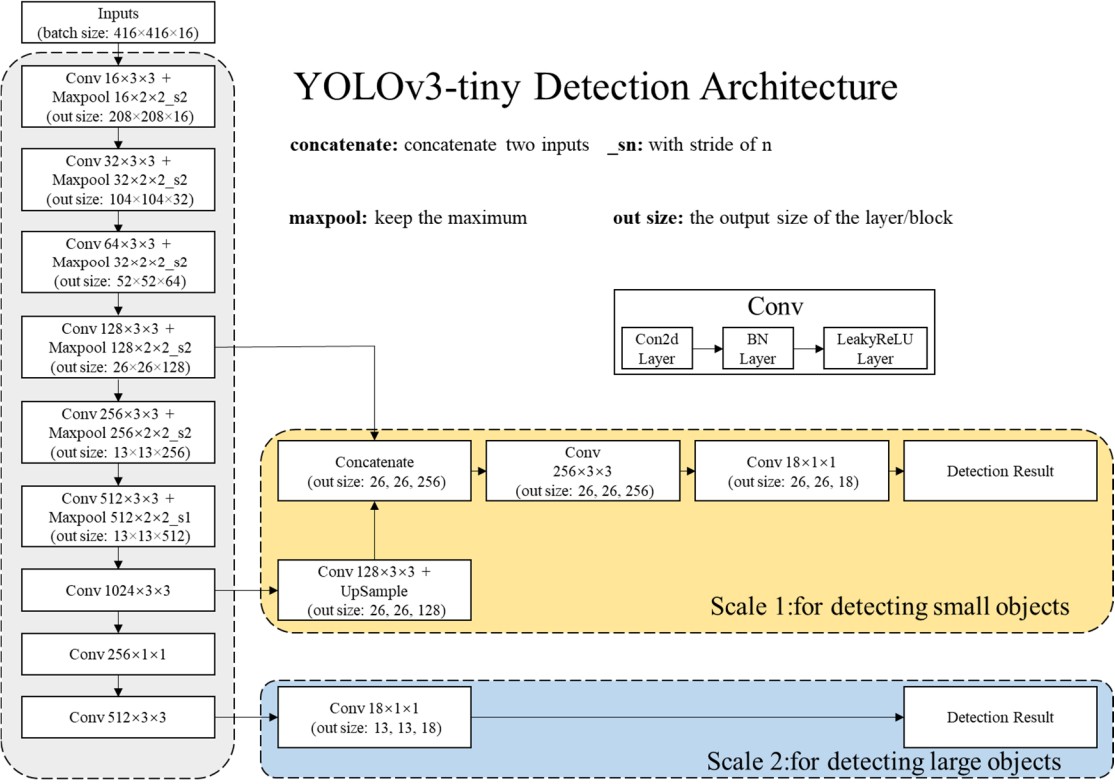

**Figure 3.** YOLOv3–tiny detection architecture.

### 2.2.2. Detection Principle

Eventual detection was conducted by applying logistic activations on the feature maps to generate 3D tensors corresponding to each scale, as shown in Figure 4. The output from the detection layer has the same size properties as the previous feature map and has detection attributes along the depth channel. Each unit of the detection layer output corresponds to a $1 \times 1 \times (3 \times (4 + 1 + 1))$ voxel inside a 3D tensor, which means that one grid cell on the feature map generates three bounding boxes. Moreover, each box has four coordinate attributes, one object confidence score, and one class probability score.

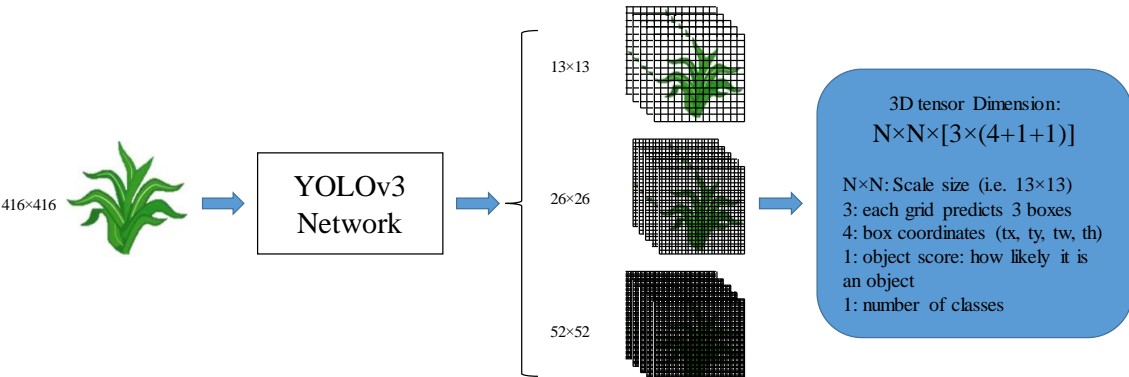

**Figure 4.** YOLOv3–tiny detection architecture.

For the prediction of bounding box coordinate attributes, three anchor priors were used for each scale. The better the quality of the priors, the easier it is for the network to conduct accurate detection. Thus, this study applied the k–means clustering method to the ground truth boxes of the training dataset to automatically identify good priors. Furthermore, unlike the Euclidean distance used in the standard k–means method, the distance metric was calculated according to Equation (17) in this study, thus eliminating the error caused by the size of the box and leading to good intersection over union (IOU) scores with the ground truth boxes:

$$d(box, centroid) = 1 - IOU(box, centroid), \tag{17}$$

There are six network output attributes for each of the bounding boxes, namely $t_x$, $t_y$, $t_w$, $t_h$, $t_o$, and $t_c$. If the cell is offset from the top left corner of the image by $(c_x, c_y)$ and the corresponding anchor prior has width and height $p_w$ and $p_h$, then the predicted bounding box attributes can be calculated by the Equations (18)–(23): $(b_x, b_y)$ denotes the position of a bounding box; $(b_w, b_h)$ denotes the width and height of a bounding box; $b_o$ and $b_c$ denote the confidence score and class probability score, respectively.

$$b_x = \sigma(t_x) + c_x, \tag{18}$$

$$b_y = \sigma(t_y) + c_y, \tag{19}$$

$$b_w = p_w e^{t_w}, \tag{20}$$

$$b_h = p_h e^{t_h}, \tag{21}$$

$$b_o = \sigma(t_o), \tag{22}$$

$$b_c = \sigma(t_c), \tag{23}$$

In reality, the number of target plants in an image is limited. As a result, a large number of inaccurate boxes exists among the detection results. To remove redundant bounding boxes, post–processing was implemented as follows.

First, a portion of the redundant boxes was removed by comparing the objectness score of a predicted bounding box with a certain threshold to reduce the computation load. If the objectness score was less than the threshold, then the object in the box was considered a non–maize plant target and the box was ignored.

Second, non–maximum suppression (NMS) was implemented to merge the remaining boxes that belonged to the same object. All boxes were sorted by the objectness scores, and the bounding boxes with the highest scores were selected. The remaining boxes were compared with the selected box. If the overlapping ratio between any remaining boxes and the selected box was greater than the IOU threshold, then those boxes were deleted from the remaining boxes. By iteratively undertaking this operation, only one valid bounding box for each target plant remained. The objectness and IOU thresholds were fine–tuned on the validation dataset and determined as 0.3 and 0.45, respectively.

2.2.3. Loss Function Definitions

The objective is to determine if maize plants are present in a scene; if so, then a bounding box is assigned to each maize plant object. Thus, the loss function consists of three parts: object confidence loss, localization loss, and bounding box size loss. This function is shown as Equation (24):

$$Loss = loss_{i,j}^o + loss_{i,j}^{xy} + loss_{i,j}^{wh}, \tag{24}$$

The object confidence loss was designed to equate the object confidence score with the IOU between the predicted bounding and ground truth boxes for the cells containing objects and make the object confidence score equal to 0 for cells containing no objects. If there are many grid cells containing no objects in an image, the object confidence scores of those cells are pushed toward 0, often overpowering the gradient from cells that do contain objects. This may cause training to diverge early and result in model instability. Therefore, $\lambda_{obj}$ and $\lambda_{noobj}$ were used as adjustment parameters to achieve a relative balance and thus avoid this problem.

$$
\text{loss}_{i,j}^{o} = \frac{\lambda_{obj}}{N^{conf}} \sum_{i=0}^{s^2} \sum_{j=0}^{B} L_{i,j}^{obj} \left( IOU_{\text{prediction}_{i,j}}^{\text{ground truth}_{i,j}} - b_o \right)^2 + \frac{\lambda_{noobj}}{N^{conf}} \sum_{i=0}^{s^2} \sum_{j=0}^{B} L_{i,j}^{noobj} (0 - b_o)^2,
\tag{25}
$$

The localization loss function penalizes localization errors of bounding boxes, and the bounding box size loss function penalizes height and width errors of bounding boxes. These were calculated by Equations (26) and (27), respectively. To ensure that small deviations in large boxes were considered to be less important than those in small boxes, the size loss function performed the square root operation on the bounding box width and height.

$$
\text{loss}_{i,j}^{xy} = \frac{\lambda_{coord}}{N_{L^{obj}}} \sum_{i=0}^{s^2} \sum_{j=0}^{B} L_{i,j}^{obj} \left[ \left( x_{i,j} - \hat{x}_{i,j} \right)^2 + \left( y_{i,j} - \hat{y}_{i,j} \right)^2 \right]
\tag{26}
$$

$$
\text{loss}_{i,j}^{wh} = \frac{\lambda_{coord}}{N_{L^{obj}}} \sum_{i=0}^{s^2} \sum_{j=0}^{B} L_{i,j}^{obj} \left[ \left( \sqrt{\omega_{i,j}} - \sqrt{\hat{\omega}_{i,j}} \right)^2 + \left( \sqrt{h_{i,j}} - \sqrt{\hat{h}_{i,j}} \right)^2 \right]
\tag{27}
$$

In Equations (25) to (27),

- $S^2$ denotes the number of grid cells in feature maps;
- B denotes the number of prediction bounding boxes of each grid cell;
- $\lambda_{coord} = 1.0$, $\lambda_{obj} = 5.0$, $\lambda_{noobj} = 1.0$;
- $N_{L^{obj}} = \sum_{i=0}^{s^2} \sum_{j=0}^{B} L_{i,j}^{obj}$;
- $N^{conf} = \sum_{i=0}^{s^2} \sum_{j=0}^{B} (L_{i,j}^{obj} + L_{i,j}^{noobj} (1 - L_{i,j}^{obj}))$;
- ground truth$_{i,j} = (x_{i,j}, y_{i,j}, w_{i,j}, h_{i,j})$;
- prediction$_{i,j} = (\hat{x}_{i,j}, \hat{y}_{i,j}, \hat{w}_{i,j}, \hat{h}_{i,j})$;
- $L_{i,j}^{obj} = \begin{cases} 1 \text{ if cell containing objects} \\ 0 \text{ else} \end{cases}$ ;
- $L_{i,j}^{noobj} = \begin{cases} 1 \text{ if cell containing no objects} \\ 0 \text{ else} \end{cases}$ .

### 2.2.4. Network Training

In the training phase, model parameters are optimized to minimize the loss function, and the mapping function is inherently learned from inputs to outputs. In this research, the training process was implemented in the Tensorflow framework on a NVIDIA Jetson Xavier platform with the following basic specifications: 512 core Volta GPU with Tensor Cores, 8 core ARM v8.2 64 bit CPU, 16 GB memory, and 32 GB storage. The GPU could only process four images of a batch size at once. The learning rate was set to be lowered

by an exponential decay as the training progressed, starting at $10^{-4}$ and decaying by 0.1 in each epoch. The training process was terminated after 40 epochs.

### 2.3. Performance Evaluation

This research focused on evaluating the detection performance from three perspectives—namely detection accuracy, robustness to various environments, and detection speed—to help select the optimal detection method for practical TF application in real fields. The main reason for the pursuit of higher accuracy was to strengthen the effect of reducing fertilizer input. Robustness performance was assessed to ensure the stability of the system in practice. Finally, fast performance is essential for on–the–go application.

In this research, only maize plants needed to be identified and located. Consequently, all the objects in an image were divided into two classes: maize and non–maize–plants. Therefore, the accuracy of target maize plant identification was evaluated by referring to the indicators in the binary classification problem. In binary classification, detection results can be divided into four types—namely true positive (TP), false positive (FP), true negative, and false negative (FN)—according to the combinations of the true and predicted classes of the detector. In this study, only TP, FP, and FN were used, because the non–maize plant areas in an image were not a concern in the research. TP means that the maize plant area in an image was correctly identified, FP means that the non–maize plant area in an image was mistakenly identified as a maize plant, and FN means that the maize plant area in an image was mistakenly identified as a non–maize plant.

The localization accuracy was evaluated by the IOU, which is calculated as Equation (28). The overlapping area between the predicted and ground truth bounding boxes is computed in the numerator, while the area of their union is computed in the denominator.

$$\text{IOU} = \frac{\text{Area of Overlap}}{\text{Area of Union}} \tag{28}$$

The detection speed was evaluated by the indicator of average frames per second (FPS), which is defined as Equation (29).

$$\text{FPS} = \frac{\text{Total images number}}{\text{Total processing time}} \tag{29}$$

The robustness of different methods was evaluated by comparing the detection performance under the changing environmental conditions mentioned previously.

All the above–mentioned methods were implemented using the Python programming language on an NVIDIA Jetson Xavier platform with the Ubuntu 18.04 system.

## 3. Experiments and Results

### 3.1. Image Data Collection

The image data used were extracted from videos taken using a GoPro5 Black camera in a maize field located at Kyoto Farm of Kyoto University, Kyoto, Japan. The data collection scenario is shown in Figure 5. The camera was held by a three–axis handheld gimbal (FEIYUTECH, G5) and mounted nearly 0.3 m above the plant canopy. The camera parameters are shown in Table 1. During video recording, the vertical resolution was set to 1080 pixels and the frame rate was set to 120 fps.

The maize was at the seedling stage, which is the key period at which fertilizer is added to promote growth in young plants. The videos were recorded under two different weather conditions, namely cloudy and sunny days, in May 2019. Images containing shadows and weed backgrounds were also considered.

To produce accurate datasets, only images containing one maize plant were selected. Image examples under different environmental cases are shown in Figure 6.

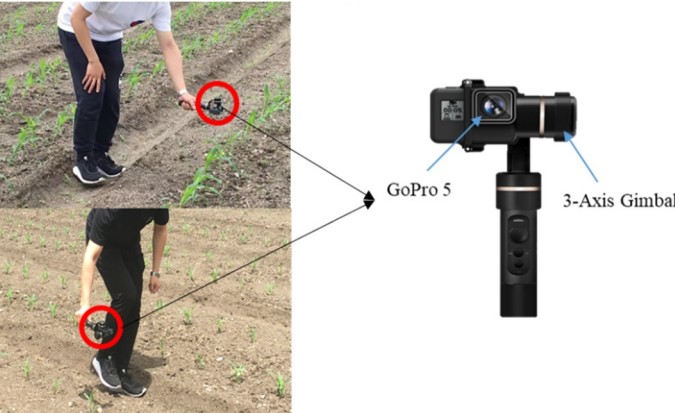

**Figure 5.** Data collection scenarios.

**Table 1.** Technical parameters of the camera.

| Parameter Types | Values |
| --- | --- |
| Video | 4K30, 1440p80, 1080p120, video stabilization |
| Connection | USB Type–C/micro–HDMI |
| Display | 2″/LCD touch display |
| Memory | Slot type: microSD/Max. slot capacity: 128 GB |
| Wireless | Bluetooth low energy/Wi–Fi: 802.11 b/g/n, 2.4 and 5 GHz |
| Location | GPS |
| Waterproof | 10 m (33 ft) |
| Battery | 1220 mAh (Removable), 4.40 V |

| (**a**) | (**b**) | (**c**) | (**d**) |
| --- | --- | --- | --- |

**Figure 6.** Image samples under different environmental cases: (**a**) cloudy days; (**b**) sunny days; (**c**) shadow background; (**d**) weed background.

The original dataset contained 820 images and was then divided into three sub–datasets: training dataset, validation dataset, and test dataset. The training dataset was prepared for training the network parameters. The validation dataset was used for fine–tuning the hyper–parameters. The test dataset was used for the performance evaluation of the methods applied in this study. Specifically, the training dataset contained 500 original images, the validation dataset contained 160 original images, and the test dataset contained 160 original images.

### 3.2. Image Data Augmentation

Agricultural environments are complex and changeable. Lighting conditions of fields vary during the day and acquired images may be unclear due to vehicle vibration. Therefore, augmentations for brightness and image definition were conducted on the original acquired images in order to simulate the real environment as much as possible. The augmentation operations are detailed in the following sections.

#### 3.2.1. Data Augmentation: Image Brightness

The brightness of the original images was adjusted via multiplication by a random factor to simulate the situations under different illumination intensities. The brightness was decreased if the factor was less than 1 and increased if the factor was greater than 1. If the image brightness is too high or low, then the edge of the target will be unclear. In the experiments, the value of the adjustment factor was limited to 0.5–2.0 because the edges of the maize plant targets can be recognized by the naked eye if the factor is in this range.

#### 3.2.2. Data Augmentation: Image Definition

The original images were randomly blurred by one of three blur operations—namely mean blur, median blur, and Gaussian blur—to simulate indistinct images. The size of the kernel used in each of the three methods was $5 \times 5$.

The augmentations were conducted on each sub–dataset. Augmented examples of each augmentation operation are shown in Figure 7. Thus, the number of samples in each sub–dataset was increased by a factor of three. As shown in Table 2, the final training dataset contained 2000 images, the final validation dataset contained 640 images, and the final test dataset contained 640 images.

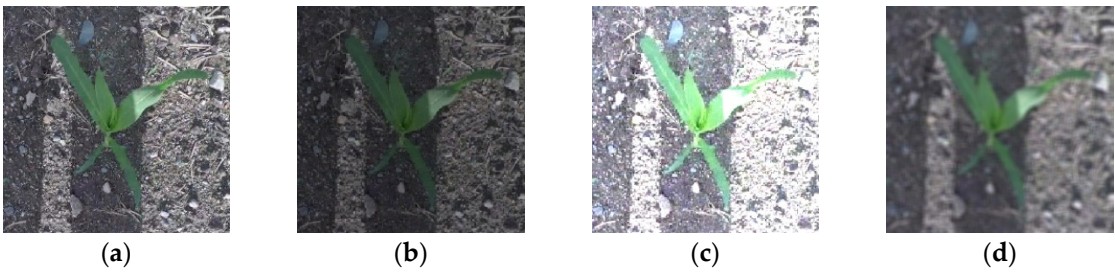

| (a) | (b) | (c) | (d) |

**Figure 7.** Image augmentations: (**a**) original image; (**b**) reduced brightness; (**c**) increased brightness; (**d**) blur processing.

**Table 2.** Generated datasets.

| Dataset Names | Operation Types | Number of Images |
| --- | --- | --- |
| Training dataset | Original | 500 |
| Training dataset | Reduced brightness | 500 |
| Training dataset | Increased brightness | 500 |
| Training dataset | Blur processing | 500 |
| Validation dataset | Original | 160 |
| Validation dataset | Reduced brightness | 160 |
| Validation dataset | Increased brightness | 160 |
| Validation dataset | Blur processing | 160 |
| Test dataset | Original | 160 |
| Test dataset | Reduced brightness | 160 |
| Test dataset | Increased brightness | 160 |
| Test dataset | Blur processing | 160 |

### 3.2.3. Image Data Annotation

Ground truths for each image sample are needed in order to train the network parameters and evaluate detection performance. The open–source graphical image annotation tool Labelme [45] was used to hand–label all the ground truth bounding boxes. The maize plant in an image was selected and labeled as the "maize" class by drawing a rectangular box around it, as shown in Figure 8. Furthermore, the annotation results were saved as JSON (JavaScript Object Notation) files for future reference.

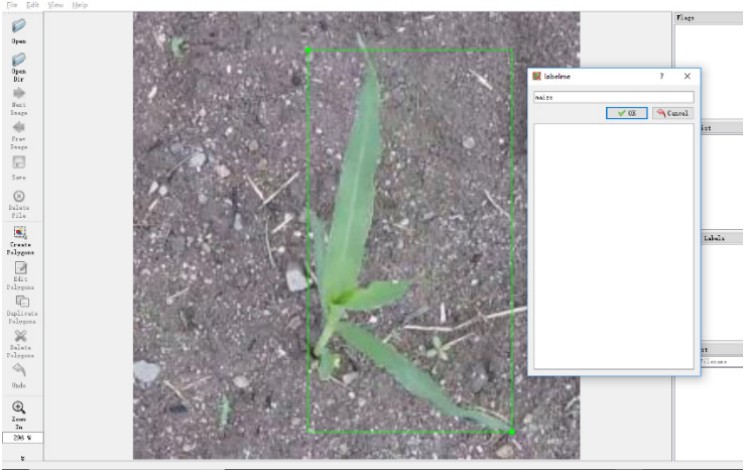

**Figure 8.** Annotation of maize plants.

### 3.3. Results and Discussion

The detection performance of the different methods was evaluated on the test dataset. First, the identification accuracy of the different methods was summarized. Second, the IOU statistics were gathered for the TP identification results of the different methods to compare the localization accuracy. Third, the robustness of the different detection methods was compared by analyzing the variation in the IOU scores across different sub–datasets. Finally, the detection speed of the different methods was calculated.

### 3.3.1. Identification Accuracy

Each image in the test dataset contained only one maize plant target. During the evaluation, the case with only one target identified in an image was considered TP, the case with more than one target identified in an image was considered FP, and the case with no targets identified in an image was considered FN. Corresponding examples of each case are shown in Figure 9.

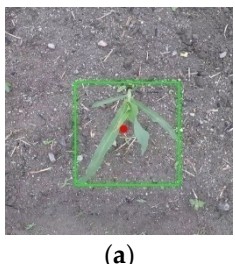 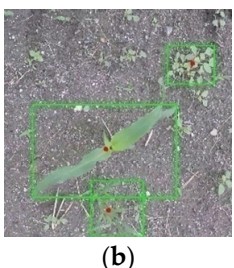 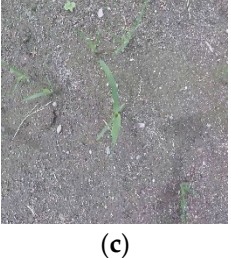

(**a**)          (**b**)          (**c**)

**Figure 9.** Samples of different identification results: (**a**) true positive (TP) sample; (**b**) false positive (FP) sample; (**c**) false negative (FN) sample.

The identification results of the test dataset using the different methods are shown in Figure 10. It can be seen that the DL methods of YOLOv3 and YOLOv3_tiny achieved higher detection accuracy levels than color index–based methods; no false detections (FP or FN) were generated by either YOLOv3 or YOLOv3_tiny. Among the accuracies for the color index–based methods, those for the ExG and ExGR–based methods were similar, with 572 and 560 TP detections, respectively. Meanwhile, the ExR and H value–based methods had lower accuracy levels, with 348 and 377 TP detections, respectively. In terms of false detections, most were FP detections, with a small number of FN detections. Specific results were as follows: the ExG–based method had 47 FP detections and 21 FN detections; the ExGR–based method had 292 FP detections and no FN detections; the ExGR–based method had 68 FP detections and 12 FN detections; the H value–based method had 263 FP detections and no FN detections.

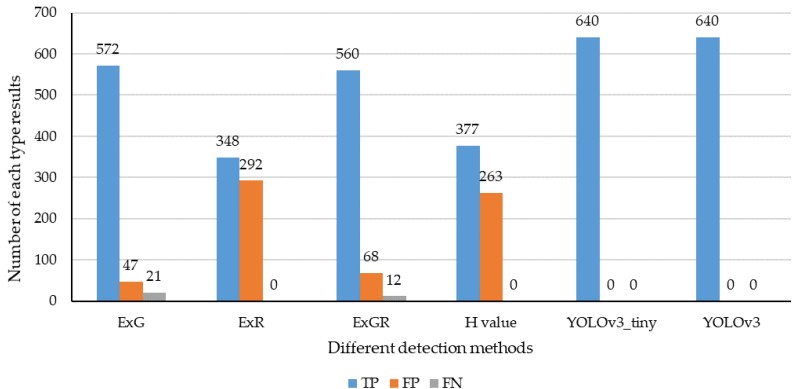

**Figure 10.** Results of different detection methods.

3.3.2. Localization Accuracy

The localization accuracy of the different methods was evaluated by the IOU score between the ground truth and predicted boxes and was statistically analyzed for only the TP results of the whole dataset. As shown in Figure 11, the IOU mean is indicated by a histogram and the standard deviation (SD) is indicated by an error bar. ExG, ExGR, and YOLOv3 presented relatively higher localization accuracy levels, with average IOUs of 0.85 ± 0.14, 0.83 ± 0.14 and 0.84 ± 0.08, respectively. Among the SDs of the three methods, that of YOLOv3 was the smallest. The average IOUs of ExR, H value, and YOLOv3_tiny were relatively smaller, at 0.59 ± 0.27, 0.64 ± 0.16, and 0.76 ± 0.09, respectively. Although the average IOU of YOLOv3_tiny was below 0.8, its SD was smaller than that of any color index–based method.

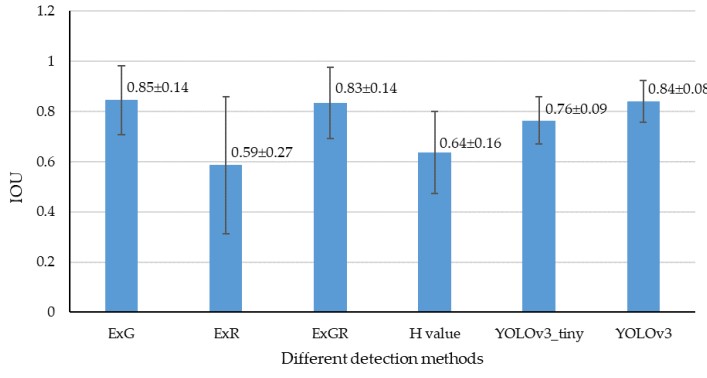

**Figure 11.** Localization accuracy comparison among different detection methods.

### 3.3.3. Robustness Analysis

Qualitative analysis was undertaken by comparing the average IOUs of each detection method across different sub–datasets—namely the original subset, blur processing subset, increased brightness subset, and reduced brightness subset—to evaluate the robustness of each detection method under various and complex field conditions. Figure 12 shows that the fluctuations in the average IOUs of the DL methods were obviously smaller than those of the color index–based methods. In terms of DL methods, the average IOUs of YOLOv3 and YOLOv3_tiny for the four different sub–datasets were in the ranges of (0.83,0.85) and (0.75,0.78) respectively, and both fluctuations were within 0.03. In terms of color index–based methods, the average IOUs of ExG, ExR, ExGR, and H value were in the ranges of (0.78,0.88), (0.38,0.78), (0.77,0.86) and (0.60,0.66), respectively. Among the color index–based methods, the H value–based method yielded the smallest fluctuation of 0.06, which was still larger than that of YOLOv3 and YOLOv3_tiny. The results demonstrate that the DL methods had smaller SDs and were more robust than the color index–based methods.

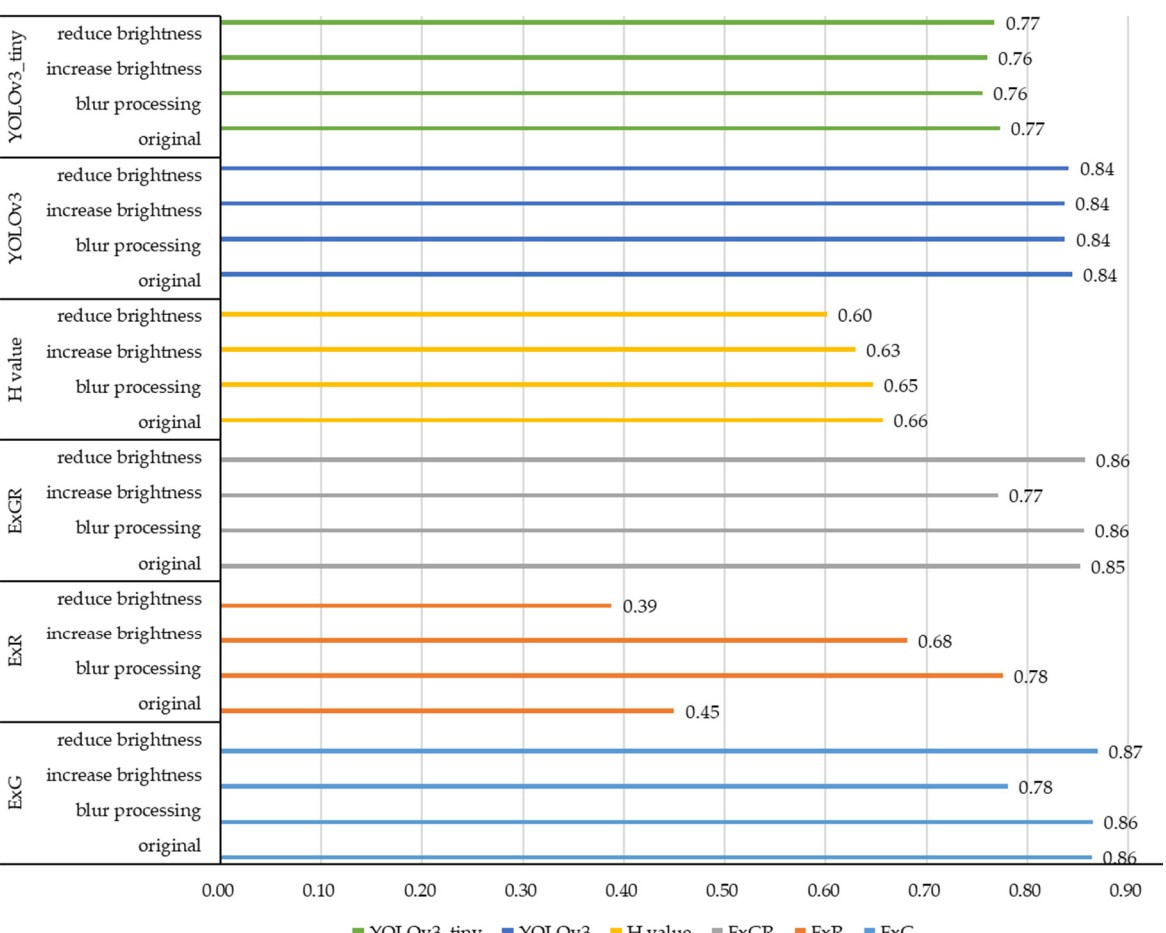

**Figure 12.** Average intersection over union (IOU) scores of each detection method across different sub–datasets, namely original image, increased brightness, decreased brightness, and blur processing.

Among the color index–based methods, the ExG and ExGR–based methods showed similar performances. Both performed relatively poorly on the increased brightness subset, with average IOUs below 0.8, although they performed well on other subsets, with average IOUs above 0.85. The ExR–based

method performed best on the blur processing subset, with an average IOU above 0.77, and it had the worst performance on the reduced brightness subset, with an average IOU below 0.4. The H value–based method exhibited more stable performance than the three other color index–based methods, with the largest average IOU of nearly 0.66 on the original subset and the smallest average IOU of around 0.60 on the reduced brightness subset. This result might be because the ExG and ExGR indices were sensitive to the strong illumination conditions, while the ExR index was relative to the redness of the soil background [46]. The H value–based methods were more robust than the other color feature–based methods because the H channel in the HSV color space could provide a separate description of the object color from the illumination content [20]. However, the largest average IOU of the H value–based method was below 0.7, which might be because the H channel was less sensitive to the greenness than the three other color indices. This result further verifies the fact that the color index–based methods are not robust and are limited to specific scenarios [13].

### 3.3.4. Detection Speed

The FPS value for each method was taken as the average of 10 repeated measurements over the whole test dataset in order to reduce errors caused by the system state, as shown in Figure 13. Overall, the detection speeds of the color feature–based methods were faster than those of the DL methods. The FPS values for all color feature–based methods were above 20, while the FPS values for YOLOv3 and YOLOv3_tiny were 6.25 and 15.3, respectively. The results can be considered reasonable because the computation of color index–based methods was much simpler than that of the DL methods, and YOLOv3_tiny used fewer convolutional layers than YOLOv3.

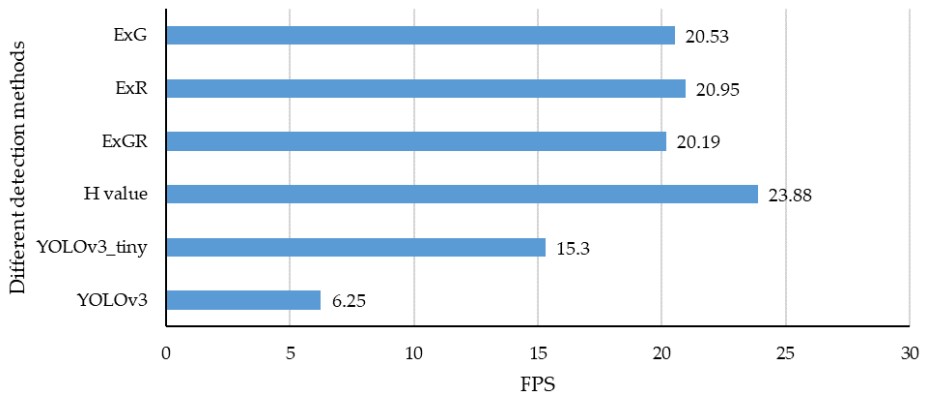

**Figure 13.** Detection speeds of different detection methods.

### 3.3.5. Discussion

Failure analysis was conducted for the color index–based methods. As observed, most of the false identifications were caused by the size threshold used in the maize plant discrimination stage. Weed plants that are bigger than the threshold value are mistaken for maize plants, resulting in FP errors. Conversely, maize plants that are smaller than the threshold value are mistaken for weeds, resulting in FN errors. This phenomenon showed that, like commonly used manually selected features, the size threshold strategy lacked generality, particularly when weeds and maize plants were a similar size.

Further investigations were conducted on the TP detection results. The results showed that the weed background also imposed difficulties for maize plant target localization for the color index–based methods, particularly when the leaves of crops and weeds overlapped. The color indices alone were not sufficient to distinguish maize plant targets from complex weed conditions because of the high degree of

similarity in their color signatures. Although combining color indices with other features such as texture and leaf venation may address such shortcomings, these features are highly dependent on the ability of experts to encode the domain knowledge. Furthermore, selected features are liable to change with different FE techniques.

To further explore why the studied DL methods had better detection accuracy than conventional color index–based methods, the first 10 layers of YOLOv3 and YOLOv3_tiny were visualized, as shown in Figures 14 and 15, respectively.

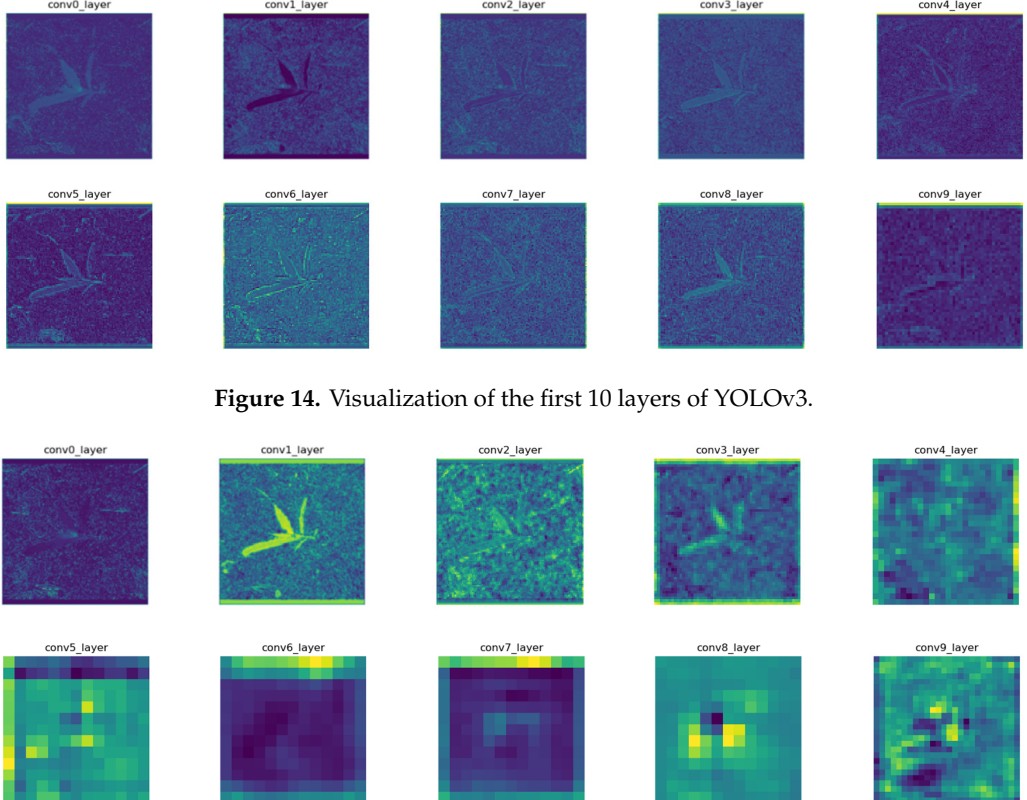

**Figure 14.** Visualization of the first 10 layers of YOLOv3.

**Figure 15.** Visualization of the first 10 layers of YOLOv3_tiny.

For YOLOv3, the features learned by the first two layers were similar to those of the color features and mainly depicted green areas in the original images. This finding showed that the first two layers of YOLOv3 tended to extract low–level features, such as colors and shapes. From layers 3 to 8, the gradient changes in leaf structures were noticeable. Thus, these layers can be viewed as a set of gradient operators that extract dedicated edges or outlines of maize plant leaves. In layer 9, only the shape–like features of the maize plant can be clearly observed, which might be interpreted as the conjunction of previous features. In layer 10, more abstract features started to be built. For YOLOv3_tiny, low–level features such as colors and shapes were only extracted by the first two layers, and high–level abstract features started to appear from the third layer.

The visualization outcomes of the network layers showed that the DL methods can extract a large number of abstract features in successive layers from the raw data. The hierarchical transformations of the feature abstractions formed representations with more discriminative power between the maize plant targets and their surroundings. Thus, they overcame the inadequacy of the methods that only used color indices.

Several examples of the detection results are shown in Figure 16 to illustrate this finding. None of the color feature–based methods were universal to all conditions, while the YOLOv3 and YOLOv3_tiny methods showed acceptable accuracies, compared to the color index–based methods, across different conditions. Thus, the advantages of the DL methods in distinguishing the discriminatory features of maize plant targets from their surroundings are clear. Furthermore, reverse engineering the DL process using deconvolution network approaches is a feasible method to efficiently extract the features that best represent the input data without the need of expert knowledge, thereby addressing the issue of ambiguity in the manual selection of features [47].

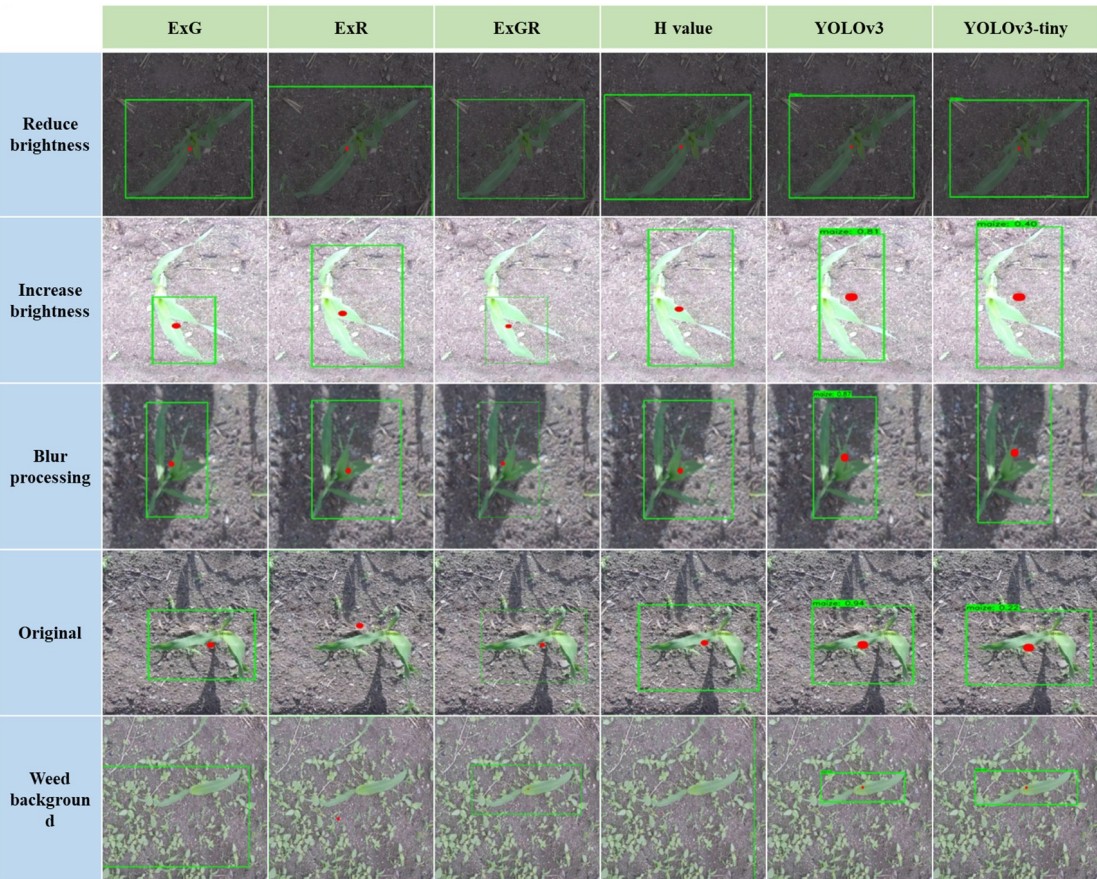

**Figure 16.** Several samples of detection results.

However, due to computational complexity, the performance of the DL methods was inferior to that of the color index–based methods in terms of detection speed. It should be noted that the detection speed will be significantly influenced by the specifications of the computer and software that are used. In addition, the target detection speed in practice should be comprehensively determined based on factors such as vehicle speed, camera installation location, lag time of subsystems, and so on. Integration of the developed detection system with a fertilization system as a basis for additional practical performance tests will therefore be carried out in the near future. In addition, network optimization strategies such as pruning techniques will also be applied to the existing DL methods, with the goal of accelerating inference on embedded devices [48]. It is expected that the detection speed will be faster on a more powerful computer or through optimized algorithm programs.

## 4. Conclusions

This study attempted to address the problem of maize plant detection. The ultimate goal was the application of target fertilization, in which fertilization is only applied to maize plant targets, thus reducing fertilization waste. The study focused on the performance evaluation of conventionally used color index–based and DL methods using several indicators—namely identification accuracy, localization accuracy, robustness to complex field conditions, and detection speed—to help make reliable detection decisions for practical application. The main conclusions are as follows.

Firstly, the color index–based methods, ExG, ExR, ExGR, and H value, have a limited ability to distinguish maize plant targets from weeds, especially when the leaves of maize plants and weeds overlap. Thus, these methods are not sufficiently robust under complex field conditions and can only be used for maize plant target detection in specific scenarios.

Secondly, compared with the color index–based methods, the two DL methods, namely YOLOv3 and YOLOv3_tiny, have higher detection accuracy levels and are more robust under complex conditions. These findings reaffirm the superiority of DL methods at automatically extrapolating and locating significant features in raw input data compared with the conventional methods. However, the detection speed of DL methods is slower than that of the color index–based methods because of their computational complexity. This problem is expected to diminish with the development of network optimization technology and improvements in computing power.

**Author Contributions:** Data collection, H.L. Methodology, H.L. Funding acquisition, H.S. and M.L. Supervision, M.I. Writing—original draft, H.L. Writing—review and editing, H.S., M.L., and M.I. All authors have read and agreed to the published version of the manuscript.

**Funding:** This research was funded by the Chinese High Technology Research and Development Research Fund (2016YFD0300600 –2016YFD0300606, 2016YFD0300600–2016YFD0300610), the National Natural Science Fund (Grant No. 31971785, 31501219), and the Fundamental Research Funds for the Central Universities (Grant No. 2020TC036).

**Acknowledgments:** The authors are grateful to Yang Li from Kyoto University for his help in neural network technology. We also appreciate the financial support of the China Scholarship Council.

**Conflicts of Interest:** The authors declare no conflict of interest.

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
