# Peer review of "Application of Color Featuring and Deep Learning in Maize Plant Detection"

_remotesensing, doi:10.3390/rs12142229_

Round 1

Reviewer 1 Report

The researchers applied two Deep Learning (DL) models for real-time classification between maize plants and other background targets in the early growing stage. The result shows that DL models have superior accuracy than that of threshold methods, while the detection speed of DL models is slower. The reviewer would like to appreciate that the application of state-of-art DL models to precision agriculture research.  Only a few comments are listed below, and the reviewer recommended a minor revision is needed without re-review. Generally, a more comprehensive investigation of the two methods would be necessary to make a better conclusion.

  1. Only a size-based filter was applied to discriminate maize and weed plants after crop segmentation for the conventional method. Would it possible to leverage other literature for a more solid conclusion about the performance of the conventional methods? There are different methods published based on shape, structure, texture. Generally, a significantly higher accuracy could be obtained by conventional ways by improving the processing protocols (i.e., preprocess before the calculation of color index) and integrating more characteristics.
  2. As the author mentioned, a balance between accuracy and detection speed is critical for real-time specific spray. Is there a target speed of the detection algorithm? For example, 100ms? Will the FPS be significantly influenced by the computer specification and the programming language used? A more comprehensive investigation is preferred.

Reviewer 2 Report

I think the manuscript is of general interest; however, there are several points that can be addressed to improve further this paper. The current study is more of proximal rather than remote sensing. As the authors argue that their goal is to help fertilization, I was wondering how the approach developed here can be scaled up with fertilization platforms? IN addition, the study only showed cases of corn in early stage of growth. What if the canopy closes and weeds exist within it? The authors also need to consider the simultaneous use of the developed approach and the fertilization platforms as the platform speed may also affect the identification accuracy. Overall, I think the current detection of maize with a handheld camera is of limited use and the authors need to address how this approach can be practically used.

Round 2

Reviewer 2 Report

No comments